# KIF2A Upregulates PI3K/AKT Signaling through Polo-like Kinase 1 (PLK1) to Affect the Proliferation and Apoptosis Levels of *Eriocheir sinensis* Spermatogenic Cells

**DOI:** 10.3390/biology13030149

**Published:** 2024-02-27

**Authors:** Yan-Shuang Zhao, Ding-Xi Liu, Fu-Qing Tan, Wan-Xi Yang

**Affiliations:** 1The Sperm Laboratory, College of Life Sciences, Zhejiang University, Hangzhou 310058, China; zhaoyanshuang2016@163.com (Y.-S.Z.); liudingxi@spermlab.org (D.-X.L.); 2The First Affiliated Hospital, College of Medicine, Zhejiang University, Hangzhou 310003, China

**Keywords:** *Eriocheir sinensis*, KIF2A, PLK1, PI3K/AKT signaling, cell proliferation, apoptosis

## Abstract

**Simple Summary:**

The kinesin superfamily is an important member of motor proteins, playing important roles in cell growth and development. Studies on KIF2A of the kinesin-13 family have mainly focused on cell division, while the relationship between KIF2A and signaling pathways remains unclear. The purpose of the present study was to determine whether KIF2A has a regulatory effect on the PI3K/AKT pathway, thus affecting spermatogenic cell proliferation and apoptosis during spermatogenesis, and whether its interacting protein PLK1 is involved in this process. RNAi experiments were performed in our study. Our results showed that KIF2A upregulates the PI3K/AKT signaling pathway through its interacting protein PLK1 during the spermatogenesis of *Eriocheir sinensis*.

**Abstract:**

*E. sinensis* is an animal model for studying the reproduction and development of crustaceans. In this study, we knocked down the *Es-Kif2a* gene by injecting dsRNA into *E. sinensis* and inhibited *Es-Plk1* gene expression by injecting PLK1 inhibitor BI6727 into *E. sinensis*. Then, the cell proliferation level, apoptosis level, and PI3K/AKT signaling expression level were detected. Our results showed that the proliferation level of spermatogenic cells decreased, while the apoptosis level increased after *Es-Kif2a* knockdown or *Es-Plk1* inhibition. In order to verify whether these changes are caused by regulating the PI3K/AKT pathway, we detected the expression of PI3K and AKT proteins after *Es-Kif2a* knockdown or *Es-Plk1* inhibition. Western Blot showed that in both the *Es-Kif2a* knockdown group and the *Es-Plk1* inhibition group, the expression of PI3K and AKT proteins decreased. In addition, immunofluorescence showed that Es-KIF2A and Es-PLK1 proteins were co-localized during *E. sinensis* spermatogenesis. To further explore the upstream and downstream relationship between Es-KIF2A and Es-PLK1, we detected the expression level of Es-PLK1 after *Es-Kif2a* knockdown as well as the expression level of Es-KIF2A after *Es-Plk1* inhibition. Western Blot showed that the expression of Es-PLK1 decreased after *Es-Kif2a* knockdown, while there was no significant change of Es-KIF2A after *Es-Plk1* inhibition, indicating that Es-PLK1 may be a downstream factor of Es-KIF2A. Taken together, these results suggest that Es-KIF2A upregulates the PI3K/AKT signaling pathway through Es-PLK1 during the spermatogenesis of *E. sinensis*, thereby affecting the proliferation and apoptosis levels of spermatogenic cells.

## 1. Introduction

The Chinese Mitten Crab, *E. sinensis*, belonging to the phylum Arthropoda, the class Crustacea, the order Decapoda, the family Trichodermaidae, the subfamily Trichoderma, and the genus *Eriocheir* [1], is one of the main aquaculture species in China. Research on *E. sinensis* reproduction is directly linked to its production. The male reproductive system of *E. sinensis* mainly includes testes, vas deferens, accessory sex glands, and ejaculatory ducts [2], among which the testis is an important place for spermatogenesis. Like mammals, spermatogenesis in *E. sinensis* also goes through the spermatogonia stage, primary spermatocyte stage, secondary spermatocyte stage, and spermatid stage (early, middle, and late) and finally develops into mature spermatozoa [3]. There are differences in cell morphology at each stage. Spermatogonia are round or oval in shape and larger than other spermatogenic cells. The chromatin in the nuclei of spermatogonia is evenly distributed. Primary spermatocytes are polygonal in shape and smaller than spermatogonia and have uneven clumps of chromatin. However, chromatin becomes dense in the nucleus of secondary spermatocytes. Early-stage spermatids are irregular in shape, and the chromatin begins to aggregate into small clumps and becomes denser. In middle-stage spermatids, the nuclei gradually become flattened and most of the chromatin is abandoned. When entering the late stage, the nuclei become more flattened and begin to curve inward. In mature spermatozoa, the nuclei become cup-shaped, called the nuclear cup (NC) [3]. Unlike the mature sperm structure of mammals, the mature spermatozoa of *E. sinensis* are noticeable, being ellipsoid-shaped, immobile, and without flagella. Due to its short spermatogenesis and fertilization time and the special structure of mature spermatozoa, *E. sinensis* is an ideal experimental animal for studying crustacean spermatogenesis. Also, research on *E. sinensis* can deepen our understanding of spermatogenesis in species that produce the non-flagellar sperm.

The successful completion of spermatogenesis is inseparable from the material and nutrient transportation of motor proteins. Our laboratory has shown that many kinesin families are involved in spermatogenesis [4]. Among them, the kinesin-13 family is the most special because kinesin-13 family members do not have the function of transporting substances; instead, their main functions are microtubule depolymerization and participating in spindle microtubule assembly, kinetochore–microtubule contact, chromosome alignment, and chromatid separation during mitosis [5,6]. In humans and mice, there are four kinesin-13 family members, including KIF2A, KIF2B, KIF2C (MCAK) and KIF24. KIF2A and KIF2B are localized at the spindle and kinetochore; MCAK can regulate spindle length and kinetochore dynamics [6]. Unexpectedly, KIF2A is also reported to be involved in the regulation of cell proliferation and apoptosis in vitro, which piqued our interest and curiosity. For example, KIF2A silencing or knockdown effectively induces apoptosis while inhibiting the proliferation, migration, and invasion of cancer cells. The effect of KIF2A on cell proliferation and apoptosis may be realized by regulating the PI3K/AKT pathway [7,8,9,10,11].

PI3K/AKT is a classic signaling pathway in cells. PI3K is composed of a regulatory subunit (p85) and a catalytic subunit (p110), and at the membrane, it converts PIP2 lipids to PIP3. PIP3 recruits two kinases, PDK1 and PDK2 (mTORC2), thereby phosphorylating AKT at Thr308 and Ser473, leading to its activation [12]. After activation, AKT acts through various substrates. AKT mediates cell growth through mTOR [12]; inhibits apoptosis and promotes cell survival through Bad [13], forkhead [14], c-Raf [15], and Caspase-9; regulates glycogen synthesis through GSK-3 [16,17]; and regulates cell cycle through GSK-3, Cyclin D1 [18], p27 Kip1 [19], and p21 Waf1/Cip1 [20]. We have reviewed that PI3K/AKT is an essential signaling for regulating the development of Sertoli cells and spermatogenic cells during spermatogenesis in mammals [21]. However, research on the PI3K/AKT pathway in crustaceans is mainly focused on the impact of environmental factors or chemicals on crustacean health through this pathway [22,23,24] and is rarely focused on spermatogenesis, let alone the relationship between the PI3K/AKT signaling and motor proteins. Therefore, this study aims to explore the role of KIF2A on spermatogenic cell proliferation and apoptosis during *E. sinensis* spermatogenesis and whether this is realized by the PI3K/AKT regulation.

In order to further improve the regulatory mechanism of KIF2A on PI3K/AKT, we also introduced a KIF2A interacting protein, polo-like kinase 1 (PLK1). PLK1 belongs to the serine/threonine kinase family and plays an important role in regulating the mitotic and meiotic cell cycle. For example, PLK1 mediates the phosphorylation of specific substrates and participates in a series of important cellular activities, such as nuclear membrane breakdown (NEBD), centrosome maturation, spindle assembly, chromosome segregation, and cytokinesis [25,26,27]. In oocytes, PLK1 promotes actin polymerization induced by polarity protein Cdc42 and promotes asymmetric oocyte division [28]. In addition, PLK1 knockdown ameliorates testicular epithelial cell apoptosis and the intestinal barrier during sepsis. PLK1 promotes autophagy and enhances lipopolysaccharide-induced apoptosis and cell permeability [29]. Interestingly, there are some studies on PLK1 and signaling pathways; for example, it has been reported that PLK1 and PI3K/AKT signaling pathways are closely related [30,31]. PLK1 may upregulate the AKT pathway by relieving the inhibitory effect of PTEN on AKT [32]. Also, PLK1 can induce autophagy by inhibiting mTORC1, which is an important downstream factor of the PI3K/AKT pathway [33].

Briefly, both KIF2A and its interacting protein PLK1 are involved in apoptosis and PI3K/AKT regulation. Thus, we speculate that these two proteins may regulate the PI3K/AKT pathway collaboratively. It is not rare for KIF2A and PLK1 to work together, as shown in previous studies. During mitosis, they co-localize at spindle microtubules and spindle poles to co-regulate mitotic progression. Their interaction depends on the Mre11-Rad50-NbS1 (MRN) complex [34]. Also, Wnt ligands can induce LRP to the DVL platform, recruit KIF2A and PLK1 to DVL, and form complexes with DVL, LRP, and the Wnt receptor Frizzled, promoting KIF2A activation and spindle pole localization [35,36]. In addition, KIF2A, PLK1, and DVL complexes are also involved in the formation of cilia [37]. However, whether/how KIF2A and PLK1 collaboratively regulate the PI3K/AKT pathway remains unclear.

In this paper, we investigate the role of KIF2A/PLK1 on spermatogenic cell proliferation and apoptosis and the PI3K/AKT pathway in the crustacean model of *E. sinensis*. Our research can not only increase knowledge of spermatogenesis in crustaceans and species with non-flagellar sperm, but also fill the gap in the regulatory mechanism of KIF2A and PLK1 on the PI3K/AKT signaling pathway. 

## 2. Materials and Methods

### 2.1. Experimental Animals

Our experimental design is shown in Figure 1. We used the Chinese mitten crabs (*E. sinensis*) as our experimental animals. Healthy 2-year-old male crabs were purchased from the Chongming Research Base of Shanghai Ocean University at a wet weight of 100 ± 10 g. All crabs were kept in plastic tanks (with a maximum of 15 crabs per tank) with clean fresh water and flowing oxygen at a stable temperature of approximately 28 °C. Before treatment, the crabs were fed and the water was changed daily for about a week until the crab population stabilized. Female ICR mice used for polyclonal antibody preparation were purchased from Slack Company (Shanghai, China). All experimental procedures and animals were approved by the Animal Experimental Ethical Inspection of the First Affiliated Hospital, College of Medicine, Zhejiang University (No. 557, 2021).

### 2.2. Cloning of Es-Kif2a and Es-Plk1 in E. sinensis

*E. sinensis* testis tissue was sent to a biological company (BGI Genomics, Beijing, China) for transcriptome library conduction. Proceed as follows: DNase digestion, rRNA removal, RNA interruption, reverse transcription, end repair, A-tail and connector addition, fragment selection and PCR enrichment, library quality testing, and Illumina sequencing. The open reading frames (ORFs) were analyzed using an online tool “https://web.expasy.org/translate/ (accessed on 6 October 2023)”. NCBI Protein BLAST “https://blast.ncbi.nlm.nih.gov/Blast.cgi (accessed on 6 October 2023)” was used to determine the CDS sequences.

Total RNA was extracted from the testes of the crabs using Trizol reagent (TaKaRa, Dalian, China). RNA reverse transcription was performed to obtain total cDNA (50 ng/μL) using PrimeScript™ RT Master Mix (for Real Time) (TaKaRa, Dalian, China). The predicted cDNA sequences of *Es-Kif2a* and *Es-Plk1* were obtained from the transcriptome library of *E. sinensis*. Based on the sequences, we designed primers using Primer Premier 5.0 software (Table 1) and conducted PCR analysis to obtain the actual coding sequence (CDS) of *Es-Kif2a* and *Es-Plk1*. The PCR protocol was 6.25 μL of 2×Flash Hot Start Master Mix (CoWin Biosciences, Beijing, China), 4.75 μL of ddH_2_O, 0.5 μL of cDNA, 0.5 μL of forward primer (10 μM), and 0.5 μL of reverse primer (10 μM) (Generay Biotech, Shanghai, China). The PCR conditions were 98 °C for 5 s, 55 °C for 10 s, and 72 °C for 15 s, repeated for 35 cycles, and 4 °C forever. The acquired PCR product was purified by the SanPrep column DNA gel extraction kit (Sangon Biotech, Shanghai, China) and later ligated to the pMD19T vector (Takara, Dalian, China) for 3~5 h. The cDNA-pMD19T mix was transformed into *E. coli* DH5α Competent Cells (TaKaRa, Dalian, China), then applied to solid LB Broth Agar medium (Sangon Biotech, Shanghai, China), and cultured at 37 °C overnight. Single colonies were selected and cultured in liquid LB Broth medium (Sangon Biotech, Shanghai, China) at 37 °C for more than 5 h. Then, the bacterial solution was sent for sequencing (BGI Genomics, Beijing, China), and the CDS of *Es-Kif2a* and *Es-Plk1* was submitted to NCBI with the Genbank IDs of ON014748 and OQ851420, respectively.

### 2.3. Quantitative Real-Time PCR (qPCR)

The transcriptional levels of *Es-Kif2a*, *Es-Pcna*, *Es-Cdk2*, *Es-Pi3k*, and *Es-Akt* were determined by quantitative real-time PCR (qPCR). The primers were designed by Primer Premier 5.0 software (Table 1). The PCR reaction solution (1×) contained 12.5 μL TB Green Premix Ex Taq (Tli RNaseH Plus) (2×) (TaKaRa, Dalian, China), 0.5 μL Forward Primer (10 μM), 0.5 μL reverse primer (10 μM), crab testis cDNA (50 ng/μL), and sterile water. We applied the CFX96 Real-Time PCR Detection System (Bio-Rad, Hercules CA, USA) to run the 3-Step PCR program: 95 °C for 5 s, 55 °C for 30 s, 72 °C for 30 s, repeated for 40 cycles.

### 2.4. Antibody

Anti-PLK1 rabbit polyclonal antibody for crab proteins was purchased (Sangon Biotech, Shanghai, China), while other polyclonal antibodies for crab proteins were prepared in the lab. The primers were designed by Primer Premier 5.0 software (Table 1) based on the CDS of these genes. The forward primers were designed with a 15~20 bp homologous sequence of pET28a vector as well as restriction endonuclease BamHI (TaKaRa, Dalian, China) recognition sites on the 5′ ends. The reverse primers were designed with a 15~20 bp homologous sequence of pET28a vector as well as restriction endonuclease EcoRI (TaKaRa, Dalian, China) recognition sites on the 5′ ends. Then, pairs of primers were used to perform PCR reactions: 98 °C for 5 s, 55 °C for 10 s, and 72 °C for 15 s, repeated for 35 cycles. At the same time, a pET28a vector (Novagen, Beijing, China) was cut by BamHI and EcoRI to be linearized. Then, the PCR products were linked to the linearized pET28a vector using ClonExpress^®^ II One Step Cloning Kit (Vazyme, Nanjing, China). The recombinant plasmids were transformed into *E. coli* DH5α Competent Cells and *E. coli* BL21 Competent Cells (TaKaRa, Dalian, China), respectively. The BL21 cells were cultured in LB medium with kanamycin (30 ng/μL) at 37 °C. When the OD600 reached 0.4~0.6, IPTG (1 mM) was added to induce target protein expression overnight. The bacterial precipitate was obtained after centrifugation and dissolved in PBS solution (1×). Then, the target proteins were extracted by ultrasonication. After centrifugation, the precipitation was collected and dissolved in 8 M urea. The target proteins were purified using the His-tagged Protein Purification Kit (Inclusion Body Protein) (CoWin Biosciences, Cambridge, MA, USA). The purified target proteins were injected into the abdominal cavity of the ICR female mice once a week, for four weeks. The first injection was injected with complete Freund’s adjuvant (Sigma-Aldrich, St. Louis, MO, USA), while the other three injections were injected with incomplete Freund’s adjuvant (Sigma-Aldrich, St. Louis, MO, USA). One week after the last injection, blood was sampled from the eyes of the mice. Then, the mice were sacrificed by cervical dislocation, and the blood samples were put into a 4 °C refrigerator overnight to separate the serum from the plasma. Then, the blood sample was centrifuged at 12,000× *g* for 20 min to obtain the serum, which contained the polyclonal antibodies of target proteins, and the serum was stored at −40 °C.

### 2.5. Western Blotting

Total protein was extracted from the crab testes placed in the RIPA lysis buffer (Strong) (Beyotime Biotechnology, Shanghai, China) with protease inhibitors using T 10 basic ULTRA-TURRAX (IKA). The tissue homogenate was centrifuged at 12,000× *g* for 15 min when the supernatant was collected. SDS solution was added to the supernatant and heated in a metal bath (100 °C, 10 min). The protein samples and protein markers (Thermo Scientific, Waltham, MA, USA) were loaded on SDS-PAGE gels and separated. Then, proteins in the gels were transferred to PVDF membranes (Merck Millipore, Burlington, MA, USA) (200 mA, 80 min). The membranes were blocked with 5% (*w*/*v*) non-fat milk (Sangon Biotech, Shanghai, China) -PBST buffer (0.05% Tween-20 in 1× PBS, *v*/*v*, pH 7.2~7.4) for 2 h. Then, the membranes were washed with PBST buffer (15~30 min) and incubated with the primary antibodies (1:1000~1:5000) overnight (4 °C). β-Actin was used as a reference gene. The membranes were washed three times with PBST buffer (15 min each time) and incubated with the corresponding secondary antibody (1:5000) at room temperature for 1 h. We washed the membranes three times with PBST buffer (15 min each time). The results were detected by a fluorescent chemiluminescence imaging machine (Tanon, Shanghai, China).

### 2.6. Immunofluorescence

Crab testes were fixed in 4% PFA (*w*/*v*) overnight, and paraffin sections were prepared by the company (Haokebio, Hangzhou, China). The sections were deparaffinized with xylene three times (2 min each time) and then hydrated with 100%, 95%, and 70% ethanol twice, respectively (2 min each time). Sections were placed in 60 mL preheated sodium citrate solution (10 mM, 95 °C) for 10 min for antigen repair. After the slices cooled naturally, the sections were washed three times with 1× PBS buffer (3 min each time). Sections were sucked dry and blocked with 1% BSA solution at room temperature for 30 min. Next, both control and treatment group sections were incubated with primary antibody (1:50) at room temperature for 1 h. The sections were washed three times with 1× PBS buffer (3 min each time). Then, the sections were incubated with a secondary antibody (1:500) at room temperature for 1 h. Nuclei were stained with DAPI (Beyotime Biotechnology, Shanghai, China) at room temperature for 20 min. Sections were sealed with an Antifade Mounting Medium (Beyotime Biotechnology, Shanghai, China). The fluorescence signals were observed by a laser scanning confocal microscope (Carl Zeiss, FV3000, Jena, Germany).

### 2.7. RNA Interference of Es-Kif2a

Primers were designed by Primer Premier 5.0 software (Table 1). The forward primers were designed with a 15~20 bp homologous sequence of the L4440 vector as well as the restriction endonuclease XbaI (TaKaRa, Dalian, China) recognition sites on the 5′ ends. The reverse primers were designed with a 15~20 bp homologous sequence of the L4440 vector as well as the restriction endonuclease Kpn I (TaKaRa, Dalian, China) recognition sites on the 5′ ends. The recombinant plasmids were transformed into HT115 competent cells (Weidi Biotechnology, Shanghai, China). IPTG (0.5 mM) was added for 5 h at 37 °C to induce dsRNA expression that specifically interfered with *Es-Kif2a* mRNA. After centrifugation, dsRNA was dissolved in 1X RNase-free PBS buffer and extracted using Trizol reagent. Crabs were randomly divided into two groups: a control group and an *Es-Kif2a* interference group (at a density of 15 crabs per experimental unit) in plastic tanks with 7 L water in each tank. In the control group, crabs were injected with 1× PBS buffer (200 μL/100 g), while in the *Es-Kif2a* interference group, crabs were injected with 200 μg dsRNA diluted with 200 μL 1× PBS buffer per 100 g body weight. The injection site was the fifth pleopod of the crab, and the injection was performed every two days. After four injections, crabs were anesthetized on the ice; then, they were sacrificed, and the testes were removed for further studies. We selected this sample size (*n* = 15) based on the mortality rate of the crabs. In general, after four injections, only half of the original crab population remained. Crabs that died before the completion of four injections were excluded.

### 2.8. Es-Plk1 Inhibition

PLK1 inhibitor Volasertib (BI 6727) was purchased (Beyotime Biotechnology, Shanghai, China) to inhibit *Es-Plk1* expression. Crabs were randomly divided into two groups: a control group and an *Es-Plk1* inhibition group (at a density of 15 crabs per experimental unit) in plastic tanks with 7 L water in each tank. In the control group, crabs were injected with 1× PBS buffer (200 μL/100 g), while in the *Es-Plk1* inhibition group, crabs were injected with 617 μg BI 6727 diluted with 200 μL 1× PBS buffer per 100 g body weight. The injection site was the fifth pleopod of the crab, and the injection was performed every two days. After four injections, the crabs were anesthetized on the ice; then, they were sacrificed, and the testes were stored at −40 °C for further studies. We selected this sample size (*n* = 15) based on the mortality rate of the crabs. In general, after four injections, only half of the original crab population remained. Crabs that died before the completion of four injections were excluded.

### 2.9. TUNEL Assay

One-Step TUNEL Apoptosis Assay Kit (Beyotime Biotechnology, Shanghai, China) was used to detect the apoptosis level of crab testes using paraffin sections after *Es-Kif2a* knockdown or *Es-Plk1* inhibition. The fluorescence signals were observed by a laser scanning confocal microscope.

### 2.10. EdU Assay

After four injections of PBS, dsRNA, or BI 6727, EdU-PBS solution was injected into crabs of all groups at a dose of 1.8 mg/100 g 2 h before sacrifice. A BeyoClick™ EdU Cell Proliferation Kit with Alexa Fluor 555 or 488 (Beyotime Biotechnology, Shanghai, China) was used to detect the cell proliferation level of crab testes after *Es-Kif2a* knockdown or *Es-Plk1* inhibition.

### 2.11. Statistical Analysis

The integrated density of Western Blot bands was analyzed by the software Image J 1.49v. Based on the mean values of the biological replicates (n ≥ 3), significant differences were analyzed in the software GraphPad Prism 8.0 by unpaired two-tailed Student’s *t*-test and expressed as means ± standard error (SEM). GraphPad Prism 8.0 was also used to output the statistical graphs.

## 3. Results

### 3.1. Localization of Es-KIF2A Protein at Different Stages of Spermatogenesis in E. sinensis

After obtaining the Es-KIF2A antibody, we applied immunofluorescence to detect the localization of Es-KIF2A during different stages of *E. sinensis* spermatogenesis. We observed that Es-KIF2A was expressed in the cytoplasm of spermatogonia, spermatocytes, early-stage spermatids, and mid-stage spermatids, but was not expressed in mature spermatozoa (Figure 2). We speculated that it is related to the growth, development, or certain functions of these spermatogenic cells. 

### 3.2. Es-Kif2a Knockdown Reduced Cell Proliferation Levels and Enhanced Apoptosis Levels In Vivo

To identify the function of *Es-Kif2a*, we interfered with the *Es-Kif2a* gene by injecting *Es-Kif2a* dsRNA into crabs. The expression of *Es-Kif2a* was detected by qPCR and Western Blot after *Es-Kif2a* interference. The results showed that the transcriptional and expression levels of *Es-Kif2a* decreased significantly compared to the control group (Figure 3A), indicating a significant knockdown effect of the *Es-Kif2a* gene. To test whether *Es-Kif2a* affects cell proliferation and apoptosis during normal spermatogenesis, we examined the level of cell proliferation and apoptosis after *Es-Kif2a* knockdown. First, we labeled *E. sinensis* testes with EdU, an analog of thymidine, which is incorporated into the double-strand DNA during DNA synthesis and reflects the level of cell proliferation. EdU assay showed that fewer EdU-555 signals could be observed after *Es-Kif2a* knockdown (Figure 3B). On the contrary, the TUNEL assay showed that many more Cy3 signals could be observed after *Es-Kif2a* knockdown (Figure 3D), indicating a reduced proliferation level but an enhanced apoptosis level after *Es-Kif2a* knockdown. Western Blot also confirmed this result. We detected the expression level of PCNA and CDK2 and found that both of them decreased significantly, while the expression level of pro-apoptotic proteins Bax and Caspase-3 increased significantly and anti-apoptotic protein Bcl-2 decreased significantly (Figure 3C,E) after *Es-Kif2a* knockdown. From these experiments, we concluded that *Es-Kif2a* knockdown led to a decreased level of cell proliferation, while the apoptosis level increased.

### 3.3. The Effects of Es-Kif2a Knockdown on PI3K/AKT Signaling

It has been reported that KIF2A affects cell proliferation and apoptosis by regulating the PI3K/AKT signaling pathway. In order to verify whether the effects of Es-KIF2A on the proliferation and apoptosis of spermatogenic cells during normal spermatogenesis are mediated by the regulation of the PI3K/AKT signaling pathway, we used Western Blot to detect the expression levels of PI3K/AKT signaling pathway proteins. The results showed that the expression levels of Es-PI3K and Es-AKT proteins decreased significantly after *Es-Kif2a* knockdown (Figure 4).

### 3.4. Co-Localization of Es-KIF2A and Es-PLK1 at Different Stages of Spermatogenesis in E. sinensis

Next, we introduced a new protein PLK1, which is a KIF2A interacting protein. We used immunofluorescence to detect the co-localization of Es-KIF2A and Es-PLK1 at different stages of spermatogenesis in *Eriocheir sinensis*. The results showed that Es-KIF2A and Es-PLK1 proteins were co-localized in spermatogonia, spermatocytes, mid-stage spermatids, and late-stage spermatids, but not in early-stage spermatids and mature spermatozoa (Figure 5).

### 3.5. Es-Plk1 Inhibition Reduced Cell Proliferation Level and Enhanced Apoptosis Level In Vivo

In order to verify whether the same phenomenon as *Es-Kif2a* knockdown will occur after *Es-Plk1* inhibition, we injected crabs with *Es-Plk1* inhibitor BI6727. Western Blot results showed that the protein expression level of *Es-Plk1* in the inhibition group (BI 6727) decreased significantly compared to the control group (Figure 6A). Western Blotting showed that the expression level of PCNA and CDK2 decreased significantly (Figure 6B), while the expression level of pro-apoptotic proteins Bax and Caspase-3 increased significantly, and anti-apoptotic protein Bcl-2 decreased significantly (Figure 6D). EdU assay showed that fewer EdU-555 signals could be observed after *Es-Plk1* inhibition (Figure 6C). On the contrary, the TUNEL assay showed that many more Cy3 signals could be observed after *Es-Plk1* inhibition (Figure 6E), indicating a reduced proliferation level but an enhanced apoptosis level after *Es-Plk1* inhibition. In conclusion, *Es-Plk1* inhibition led to a decreased level of cell proliferation, but to an increased level of apoptosis, which is consistent with *Es-Kif2a* knockdown.

### 3.6. The Effects of Es-Plk1 Inhibition on PI3K/AKT Signaling

It has been reported that PLK1 affects cell proliferation and apoptosis by regulating the PI3K/AKT signaling pathway [32]. In order to verify whether the effect of Es-PLK1 on the proliferation and apoptosis of spermatogenic cells during normal spermatogenesis is mediated by the regulation of the PI3K/AKT signaling pathway, we used Western Blot to detect the expression levels of PI3K, AKT proteins. The results showed that the expression levels of Es-PI3K and Es-AKT proteins decreased significantly after *Es-Plk1* inhibition (Figure 7). 

### 3.7. Es-PLK1 Is a Downstream Factor of Es-KIF2A

Through the above experiments, we found that both Es-KIF2A and Es-PLK1 can affect the proliferation and apoptosis of spermatogenic cells by regulating the PI3K/AKT signaling pathway. To figure out the upstream and downstream relationship between Es-KIF2A and Es-PLK1, we examined the expression level of Es-PLK1 protein after *Es-Kif2a* knockdown and the expression level of Es-KIF2A protein after in-vivo *Es-Plk1* inhibition. Western Blotting showed that the expression level of Es-PLK1 protein decreased significantly after *Es-Kif2a* knockdown (Figure 8A), while there was no significant change in the expression level of Es-KIF2A protein after *Es-Plk1* inhibition (Figure 8B).

## 4. Discussion and Perspectives

### 4.1. Es-KIF2A and Es-PLK1 Are Both Involved in Maintaining the Proliferation of Spermatogenic Cells and Inhibiting the Apoptosis of Spermatogenic Cells

Previous studies on KIF2A and PLK1 mainly focused on cell division processes. In this study, we first investigated their roles in spermatogenic cell proliferation and apoptosis in vivo. For the first time, we cloned the *Es-Kif2a* and *Es-Plk1* genes and performed multiple amino acid sequence alignment and evolutionary tree analysis on them (Appendix A). These results showed that Es-KIF2A and Es-PLK1 are very conserved evolutionarily, especially when compared to species in the same genus of Crustacea, laying the foundation for the study of crustaceans. Next, Western Blot and EdU/TUNEL staining were performed to detect spermatogenic cell proliferation and apoptosis after *Es-Kif2a* knockdown or *Es-Plk1* inhibition. Our results showed that both *Es-Kif2a* and *Es-Plk1* can affect spermatogenic cell proliferation and apoptosis during *E. sinensis* spermatogenesis. 

However, in the future, more methods will be needed to detect the expression of other proliferation/apoptosis-related proteins, and further validation in in vitro cell lines is necessary.

### 4.2. Effects of Es-KIF2A and Es-PLK1 on the Proliferation and Apoptosis of Spermatogenic Cells Are Realized by Regulating the PI3K/AKT Signaling Pathway

Next, our Western Blot results showed that the expression levels of PI3K and AKT proteins decreased significantly after *Es-Kif2a* knockdown or *Es-Plk1* inhibition, indicating that they have a positive regulatory mechanism on PI3K/AKT. AKT activation, on the one hand, promotes the expression of proliferating proteins PCNA, CDK2, and anti-apoptotic protein Bcl-2 and on the other hand, inhibits the expression of pro-apoptotic proteins Bax and Caspase-3 (Figure 9). Thus, our findings demonstrated that KIF2A/PLK1 affects cell proliferation and apoptosis through the PI3K/AKT pathway. Moreover, the immunofluorescence results showed that Es-KIF2A and Es-PLK1 co-localized during different stages of *E. sinensis* spermatogenesis, which demonstrates that KIF2A and PLK1 may coordinate to regulate the PI3K/AKT pathway. These results fill the gap in the regulation of the PI3K/AKT pathway by these two genes.

As we mentioned before, the MRN complex [34] and DVL ligand [35,36,37] are responsible for KIF2A-PLK1 complex recruitment during mitosis or cilia formation. In this case, are MRN and DVL also involved in the recruitment of the KIF2A/PLK1 complex during the regulation of the PI3K/AKT pathway? Further research is needed to answer this question (Figure 9).

Another question requires further research: how does the KIF2A-PLK1 complex regulate the PI3K/AKT pathway? As we mentioned earlier, PLK1 may upregulate the AKT pathway by relieving the inhibitory effect of PTEN on AKT [32]. In addition, we speculate that some mitotic regulatory factors may be involved in this process (Figure 9). During mitosis, PLK1 interacts and phosphorylates the N-terminus of SPAG5 (also known as astrin) [38,39,40], which has been reported to upregulate the PI3K/AKT pathway in cancer cells in vitro [41,42,43,44] through the centrosome protein CEP55 [45]. In *Xenopus*, Gwl (MASTL in humans) also interacts with and is phosphorylated by PLK1 during the cell cycle [46]. MASTL-PP2A is a well-known cell cycle checkpoint system, which restrains PI3K/AKT activity during the human cell cycle [47,48]. These speculations all require further research.

Furthermore, the following questions require further exploration. Can KIF2A/PLK1 also regulate other signaling pathways related to cell proliferation and apoptosis, like TGF-β, Wnt-β-catenin, MAPK/ERK, JNK, NF-κB, etc.? And do these pathways directly interact with each other through crossover? This is shown in Figure 9.

### 4.3. Es-PLK1 Is a Downstream Factor of Es-KIF2A

At the end of this study, we preliminarily explored the upstream and downstream relationships between Es-KIF2A and Es-PLK1. Western Blot showed that the expression level of the Es-PLK1 protein decreased significantly after *Es-Kif2a* knockdown, while there was no significant change in Es-KIF2A expression after *Es-Plk1* inhibition. Thus, we tentatively conclude that Es-PLK1 is a downstream factor of Es-KIF2A. 

In the future, more experiments, such as subcellular co-localization, co-immunoprecipitation (co-IP), interaction site/domain analysis, etc., will need to be performed.

### 4.4. Effects of Es-Kif2a Knockdown or Es-Plk1 Inhibition on Testicular Somatic Cells of E. sinensis

Our fluorescence images focused on spermatogenic cells and confirmed the effect of *Es-Kif2a* knockdown or *Es-Plk1* inhibition on spermatogenic cell proliferation and apoptosis. However, in addition to germ cells, *E. sinensis* testes also contain somatic cells, such as Sertoli cells, Leydig cells, and myoid cells, which constitute the microenvironment or the niche of testes and are essential for regulating normal spermatogenesis. Whether *Es-Kif2a* knockdown or *Es-Plk1* inhibition also affects these somatic cells requires further research of in vitro cell lines. 

## 5. Conclusions

In conclusion, our results showed that Es-KIF2A upregulates the PI3K/AKT signaling pathway through Es-PLK1 during *E. sinensis* spermatogenesis, thus affecting the proliferation and apoptosis levels of spermatogenic cells. However, further research needs to be conducted to verify these results.

## Figures and Tables

**Figure 1 biology-13-00149-f001:**
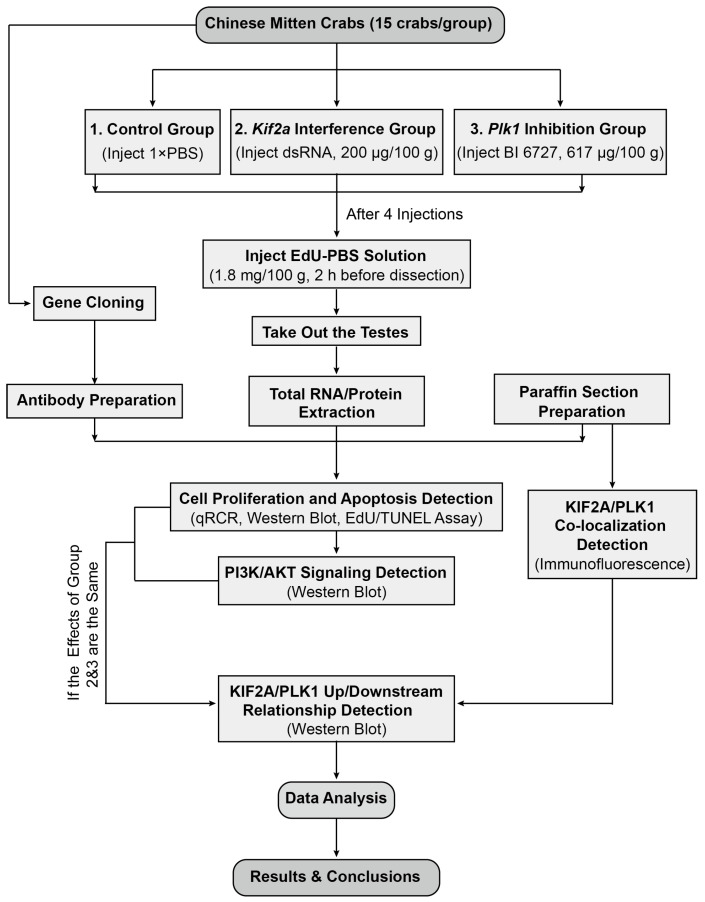
The overall experimental design of our study. See text for details.

**Figure 2 biology-13-00149-f002:**
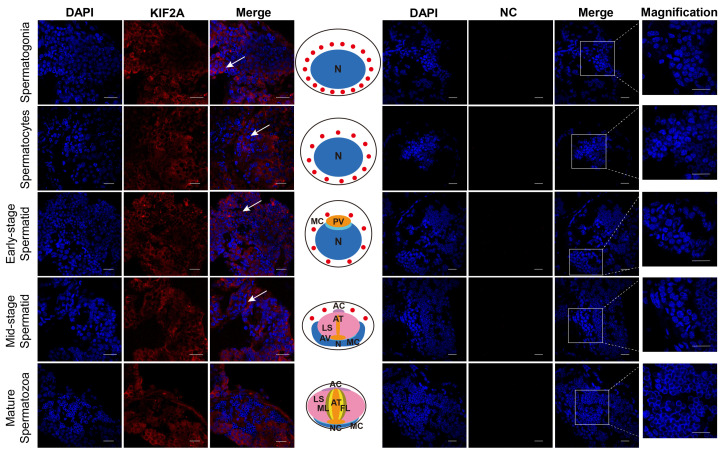
Location of the KIF2A protein during different stages of spermatogenesis in *E. sinensis*. The blue signal represents nuclei stained with DAPI; the red signal shows the distribution of Es-KIF2A protein; the white arrows represent the Merge signals. NC refers to the negative control group, which did not incubate the primary antibody, and no signals other than DAPI were detected. From top to bottom, the spermatogonia stage, spermatocyte stage, early-stage spermatids, mid-stage spermatids, and mature spermatozoa are shown. In the middle of this figure, the pattern diagram according to the results is shown. N: nucleus; MC: membrane complex; PV: pre-acrosomal vesicles; AC: acrosome cap; AT: acrosome tube; LS: sheet layer structure; AV: acrosome vesicle; ML: intermediate layer; FL: fibrous layer; NC: nuclear cup. Scale bar: 20 μm.

**Figure 3 biology-13-00149-f003:**
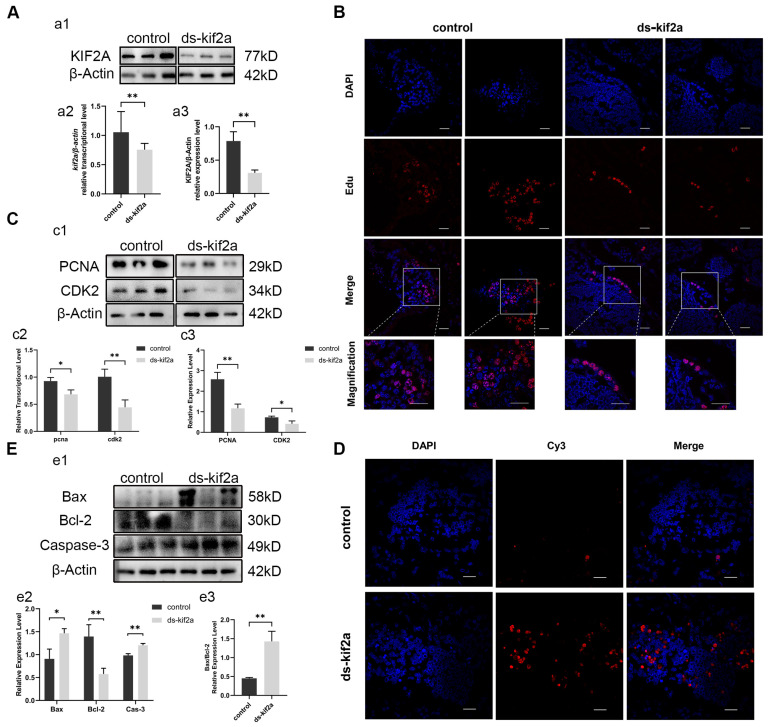
*Es-Kif2a* interference reduced spermatogenic cell proliferation and enhanced apoptosis level in the testes of *E. sinensis*. The control group was injected with 1× PBS solution, while the interference group (ds-Kif2a) was injected with equal dsRNA against *Es-Kif2a*. (**A**) *Es-Kif2a* interference effect detection. (**a1**) Western Blot detection of Es-KIF2A expression. (**a2**) Quantification analysis of transcriptional level of *Es-Kif2a* (n = 13, −0.2985 ± 0.1024, *p* = 0.0076). (**a3**) Quantification analysis of expression level of Es-KIF2A (n = 3, −0.4778 ± 0.08437, *p* = 0.0048). (**B**) Immunofluorescence detection of EdU cell proliferation signals. The blue signal represents the nucleus stained with DAPI, the red signal represents Edu-Alexa Fluor 555, and the Merge signal represents the fusion signal of the two. Scale bar: 20 μm. (**C**) Detection of proliferation-related gene expression. (**c1**) Western Blot detection of Es-PCNA, Es-CDK2 expression. (**c2**) Quantification analysis of transcriptional levels of *Es-Pcna* (n = 3, −0.2433 ± 0.06074, *p* = 0.0160) and *Es-Cdk2* (n = 3, −0.5633 ± 0.1129, *p* = 0.0076). (**c3**) Quantification analysis of expression levels of Es-PCNA (n = 3, −1.413 ± 0.2269, *p* = 0.0034) and Es-CDK2 (n = 3, −0.3185 ± 0.08825, *p* = 0.0226). (**D**) Immunofluorescence detection of TUNEL apoptosis signals. The blue signal represents the nucleus stained with DAPI, the red signal represents Cy3, and the Merge signal represents the fusion signal of the two. Scale bar: 20 μm. (**E**) Detection of apoptosis-related proteins. (**e1**) Western Blot detection of Es-Bax, Es-Bcl-2, Es-Caspase-3 expression. (**e2**) Quantification analysis of expression levels of Es-Bax (n = 3, 0.5584 ± 0.1364, *p* = 0.0149), Es-Bcl-2 (n = 3, −0.8206 ± 0.1657, *p* = 0.0078), and Es-Caspase-3 (n = 3, 0.2217 ± 0.03205, *p* = 0.0023). (**e3**) Quantification analysis of Es-Bax/Es-Bcl-2 ratio (n = 3, 0.9760 ± 0.1549, *p* = 0.0032). The integrated density was analyzed by the software Image J, qPCR results were analyzed by Microsoft Excel, and the analyzed data were imported into GraphPad Prism 8 software for statistical analysis. The significant differences were analyzed using the unpaired Student’s *t*-test and expressed as mean ± SEM (n ≥ 3). “*” above the columns indicates *p* < 0.05, and “**” indicates *p* < 0.01.

**Figure 4 biology-13-00149-f004:**
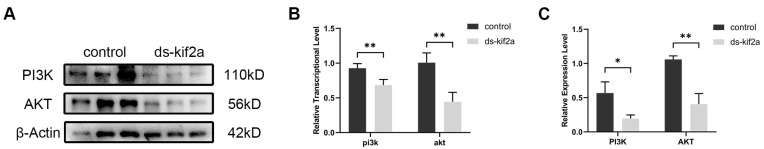
The expression levels of PI3K and AKT proteins decreased after *Es-Kif2a* knockdown in the testes of *E. sinensis*. (**A**) Western Blot detection of Es-PI3K and Es-AKT expression after *Es-Kif2a* knockdown. (**B**) Quantification analysis of transcriptional levels of *Es-Pi3k* (n = 3, −0.7667 ± 0.1400, *p* = 0.0054) and *Es-Akt* (n = 3, −0.5000 ± 0.07134, *p* = 0.0022). (**C**) Quantification analysis of expression levels of Es-PI3K (n = 3, −0.3723 ± 0.09844, *p* = 0.0194) and Es-AKT (n = 3, −0.6544 ± 0.09454, *p* = 0.0023). The integrated density was analyzed by the software Image J, and the analyzed data were imported into GraphPad Prism 8 software for statistical analysis. The significant differences were analyzed using the unpaired Student’s *t*-test and expressed as mean ± SEM (n = 3). “*” above the columns indicates *p* < 0.05, and “**” indicates *p* < 0.01.

**Figure 5 biology-13-00149-f005:**
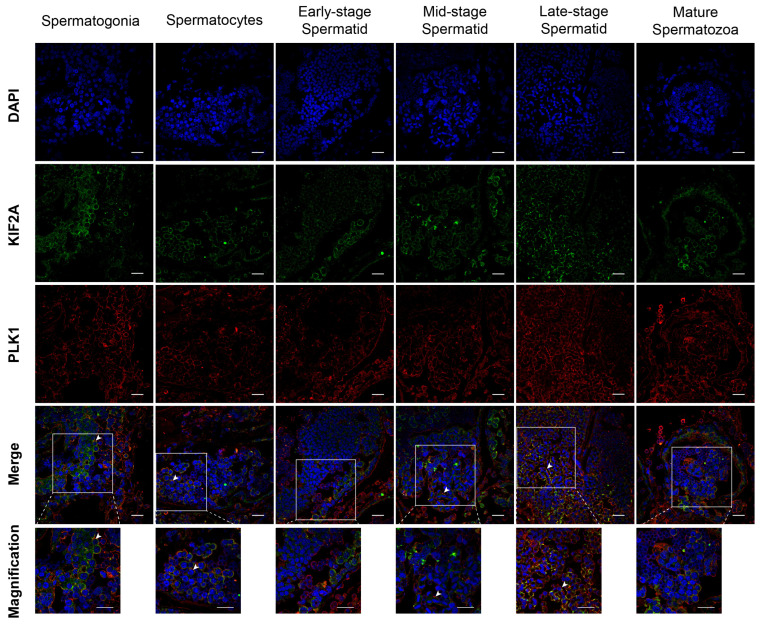
Co-localization of Es-KIF2A and Es-PLK1 during different stages of spermatogenesis in *E. sinensis*. The blue signal represents nuclei stained with DAPI; the green signal represents the distribution of the Es-KIF2A protein; the red signal represents the distribution of the Es-PLK1 protein. The white arrows show the Merge signals. From left to right, the spermatogonia stage, spermatocyte stage, early-stage spermatids, mid-stage spermatids, late-stage spermatids, and mature spermatozoa are shown. Scale bar: 20 μm.

**Figure 6 biology-13-00149-f006:**
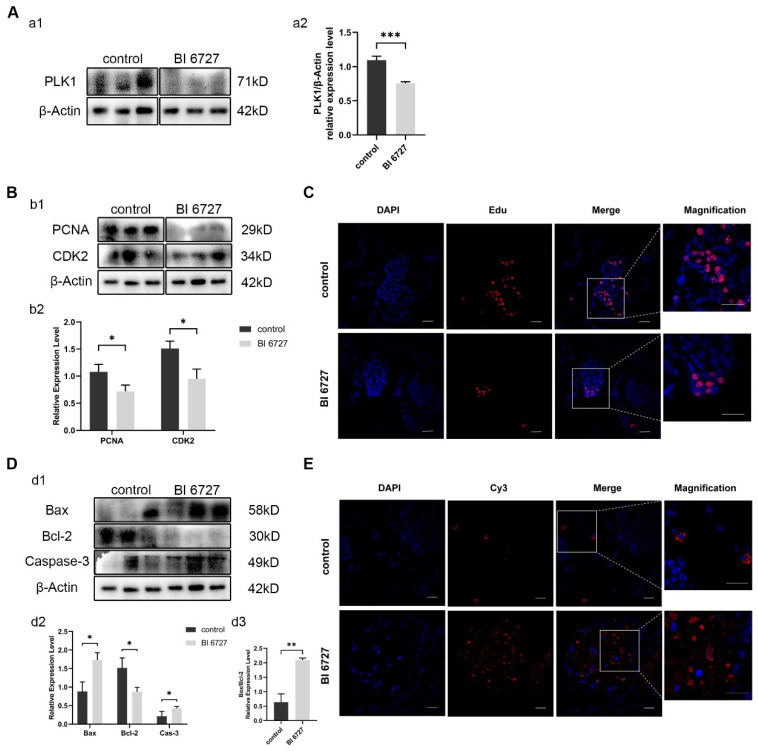
*Es-Plk1* inhibition reduced spermatogenic cell proliferation and enhanced the apoptosis level in the testes of *E. sinensis*. The control group was injected with 1× PBS solution, while the inhibition group (BI 6727) was injected with equal BI 6727 (a PLK1 inhibitor) solution. (**A**) *Es-Plk1* inhibition effect detection. (**a1**) Western Blot detection of Es-PLK1 expression. (**a2**) Quantification analysis of expression level of Es-PLK1 (n = 3, −0.3385 ± 0.03686, *p* = 0.0008). (**B**) Detection of proliferation-related gene expression. (**b1**) Western Blot detection of Es-PCNA, Es-CDK2 expression. (**b2**) Quantification analysis of expression levels of Es-PCNA (n = 3, −0.3613 ± 0.1051, *p* = 0.0264) and Es-CDK2 (n = 3, −0.5614 ± 0.1304, *p* = 0.0126). (**C**) Immunofluorescence detection of EdU cell proliferation signals. The blue signal represents the nucleus stained with DAPI, the red signal represents Edu-Alexa Fluor 555, the Merge signal represents the fusion signal of the two, and the magnification is the amplification of the Merge signal. Scale bar: 20 μm. (**D**) Detection of apoptosis-related proteins. (**d1**) Western Blot detection of Es-Bax, Es-Bcl-2, and Es-Caspase-3 expression. (**d2**) Quantification analysis of expression levels of Es-Bax (n = 3, 0.8483 ± 0.1846, *p* = 0.0101), Es-Bcl-2 (n = 3, −0.6436 ± 0.1697, *p* = 0.0192), and Es-Caspase-3 (n = 3, 0.2181 ± 0.07804, *p* = 0.0491). (**d3**) Quantification analysis of Es-Bax/Es-Bcl-2 ratio (n = 3, 1.455 ± 0.1690, *p* = 0.0010). (**E**) Immunofluorescence detection of TUNEL apoptosis signals. The blue signal represents the nucleus stained with DAPI, the red signal represents Cy3, the Merge signal represents the fusion signal of the two, and the magnification is the amplification of the Merge signal. Scale bar: 20 μm. The integrated density was analyzed by the software Image J, qPCR results were analyzed by Microsoft Excel, and the analyzed data were imported into GraphPad Prism 8 software for statistical analysis. The significant differences were analyzed using the unpaired Student’s *t*-test and expressed as mean ± SEM (n = 3). “*” above the columns indicates *p* < 0.05, and “**” indicates *p* < 0.01, “***” indicates *p* < 0.001.

**Figure 7 biology-13-00149-f007:**
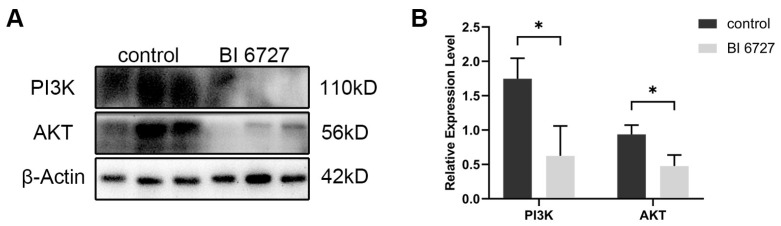
The expression level of PI3K and AKT proteins decreased after *Es-Plk1* inhibition in the testes of *E. sinensis*. (**A**) Western Blot detection of Es-PI3K and Es-AKT expression levels after *Es-Plk1* inhibition. (**B**) Quantification analysis of expression levels of Es-PI3K (n = 3, −1.122 ± 0.3040, *p* = 0.0210) and Es-AKT (n = 3, −0.4606 ± 0.1216, *p* = 0.0193). The integrated density was analyzed by the software Image J, and the analyzed data were imported into GraphPad Prism 8 software for statistical analysis. The significant differences were analyzed using the unpaired Student’s *t*-test and expressed as mean ± SEM (n = 3). “*” above the columns indicates *p* < 0.05.

**Figure 8 biology-13-00149-f008:**
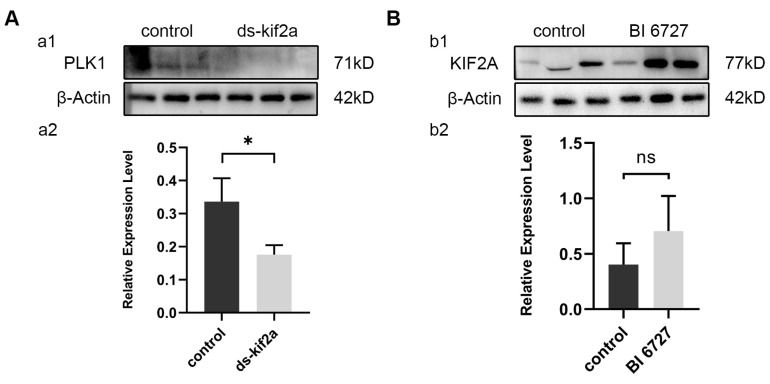
Es-PLK1 is a downstream factor of Es-KIF2A. (**A**) Es-PLK1 detection after *Es-Kif2a* knockdown. (**a1**) Western Blot detection of Es-PLK1 expression. (**a2**) Quantification analysis of expression level of Es-PLK1 (n = 3, −0.1603 ± 0.04432, *p* = 0.0224). (**B**) Es-KIF2A detection after *Es-Plk1* inhibition. (**b1**) Western Blot detection of Es-KIF2A expression. (**b2**) Quantification analysis of expression level of Es-KIF2A (n = 3, 0.3021 ± 0.2147, *p* = 0.2322). The integrated density was analyzed by the software Image J, and the analyzed data were imported into GraphPad Prism 8 software for statistical analysis. The significant differences were analyzed using the unpaired Student’s *t*-test and expressed as mean ± SEM (n = 3). “ns” above the columns indicates not statistically significant, and “*” above the columns indicates *p* < 0.05.

**Figure 9 biology-13-00149-f009:**
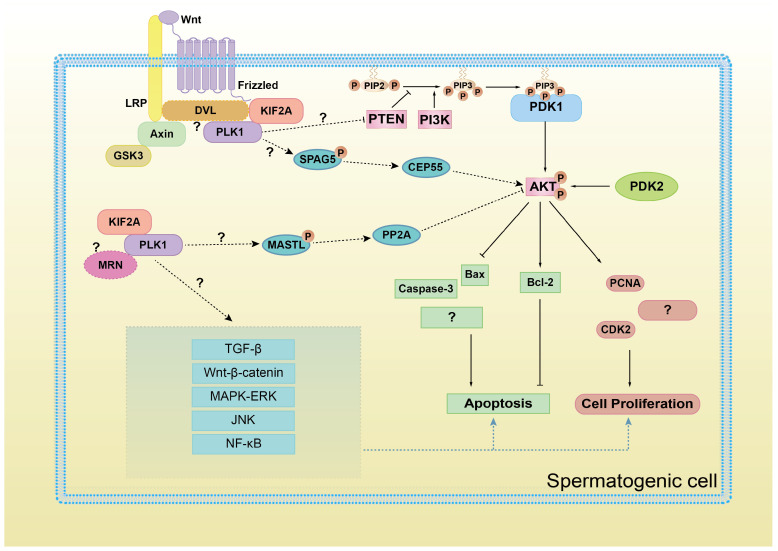
A schematic diagram of KIF2A/PLK1 complex co-regulating PI3K/AKT signaling and proliferation and apoptosis in spermatogenic cells. AKT activation, on the one hand, promotes the expression of proliferating proteins PCNA, CDK2, and anti-apoptotic protein Bcl-2 and on the other hand, inhibits the expression of pro-apoptotic proteins Bax and Caspase-3. However, whether other protein factors are involved in the regulation of PI3K/AKT pathway by KIF2A/PLK1 complex, whether KIF2A/PLK1 complex also regulate other signaling pathways, whether other cell proliferation/apoptosis proteins are also influenced, these questions still requires further study.

**Table 1 biology-13-00149-t001:** The primer sequences used in the present study.

Primer Name	Primer Sequences (5′-3′)	Purpose
*Kif2a*-CDS-F1	GACTACGTGTCCGGTCTCAACG	Fragment cloning
*Kif2a*-CDS-R1	GCCACCATAGCATAAATTCCCT	Fragment cloning
*Kif2a*-CDS-F2	AACCTGTTTTGCCTACGGTCAG	Fragment cloning
*Kif2a*-CDS-R2	ACATTGTTGGTGCCATTCTCG	Fragment cloning
*Kif2a*-CDS-F3	GACGAGAATGGCACCAACAATG	Fragment cloning
*Kif2a* CDS-R3	TCTATGTCAAGGCTGGGATGCT	Fragment cloning
*Plk1*-CDS-F1	TGCACCATGACTAGCCACGC	Fragment cloning
*Plk1*-CDS-R1	CAACCAATGGACCAGACATCAA	Fragment cloning
*Plk1*-CDS-F2	ATTGGCGACTTTGGTCTGGC	Fragment cloning
*Plk1*-CDS-R2	TCGTTGACGTTGTCCCGCTC	Fragment cloning
*Plk1*-CDS-F3	CCTGATGAGCGGGACAACGT	Fragment cloning
*Plk1*-CDS-R3	GGGAATGTCAGCATGGGGTG	Fragment cloning
*Kif2a*-q-F	CGGCATGAACTGCTGTGAAC	qPCR
*Kif2a*-q-R	TGTTGGTGCCATTCTCGTCA	qPCR
*Pcna*-q-F	ATGGACAACTCCCACGTGTC	qPCR
*Pcna*-q-R	GGCTGCACACTTCAGGATCT	qPCR
*Cdk2*-q-F	CTCCCAGACTACAAGAGCACTTTCCC	qPCR
*Cdk2*-q-R	ACAGGTTTCTCCCGGTGGCATT	qPCR
*Pi3k*-q-F	GCAAAAGCTACAAAGGCACGG	qPCR
*Pi3k*-q-R	GCATGGACTCCCCTATCTGGTC	qPCR
*Akt*-q-F	TGCCGACGGTCACATAAAGAT	qPCR
*Akt*-q-R	TAGCCCCACCAGTCAACACC	qPCR
*actin*-q-F	CGAGGCTACACCTTCACGAC	qPCR
*actin*-q-R	ACGCGGCAGTGGTCATTT	qPCR
KIF2A-Ab-F	AATGGGTCGCGGATCCCAAGGCGACAAGGCGTAT	Prokaryotic expression
KIF2A-Ab-R	GACGGAGCTCGAATTCCCACCATTCCAGATTTTCCAAC	Prokaryotic expression
Bax-Ab-F	AATGGGTCGCGGATCC CGTGCCGTTCAGGATAGCCAA	Prokaryotic expression
Bax-Ab-R	GACGGAGCTCGAATTCAAGACGTAACCGGGGAGGT	Prokaryotic expression
Bcl-2-Ab-F	CAGCAAATGGGTCGCGGATCCGATAAGGTGGTTCGCTCCGTC	Prokaryotic expression
Bcl-2-Ab-R	TTGTCGACGGAGCTCGAATTCCACGTCACCAGCTCCCCG	Prokaryotic expression
Caspase 3-Ab-F	AATGGGTCGCGGATCCATGAAGCGGCAGGAAGAGG	Prokaryotic expression
Caspase 3-Ab-R	GACGGAGCTCGAATTCGTCGAACTGGTGAGCGAAG	Prokaryotic expression
ds-*Kif2a*-F	TGGCGGCCGCTCTAGAGCAACCTGTTTTGCCTACGGTCA	DsRNA
ds-*Kif2a*-R	TATAGGGCGAATTGGGTACCCGCCTTGTCGCCTTGCTTTTCT	DsRNA

Notes: F, forward. R, reverse. Underlined, T7 promoter sequence. GenBank number: *Es-Akt* (KY412800.1), *Es-Caspase-3* (MH183147.1). Other genes have not yet been uploaded to NCBI.

## Data Availability

Data are contained within the article and Appendix A.

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
