# Peer review of "KIF2A Upregulates PI3K/AKT Signaling through Polo-like Kinase 1 (PLK1) to Affect the Proliferation and Apoptosis Levels of *Eriocheir sinensis* Spermatogenic Cells"

_biology, 2024, doi:10.3390/biology13030149_

Round 1
Reviewer 1 Report
Comments and Suggestions for Authors
The article “KIF2A upregulates PI3K/AKT signaling through polo-like ki2 nase 1 (PLK1) to affect the proliferation and apoptosis levels of 3 Eriocheir sinensis spermatogenic cells” by Zhao et al. is dealing with a very interesting topic with a vast applied potential. In this study, they knocked down the Es-Kif2a gene by injecting dsRNA into E. sinensis, and overexpressed Es-KIF2A protein in HEK293T cell lines in vitro. Then the cell proliferation level, apoptosis level and PI3K/AKT signaling expression level were detected. Similarly, they inhibited Es-Plk1 gene expression by injecting PLK1 inhibitor BI6727 into E. sinensis, and overexpressed Es-PLK1 protein in HEK293T cell lines in vitro. The cell proliferation level, apoptosis level and PI3K/AKT signaling expression level were also detected. Finally, Western blotting was conducted to explore the upstream and downstream relationship between Es-KIF2A and Es-PLK1. However, some major issues significantly compromised the quality of this MS. I suggest this article be rejected.
Abstract
The first time a species name or other abbreviation must be written in its entirety, and subsequent reoccurrences should be abbreviated. Eriocheir sinensis- (E. sinensis)
Gene names should be italicized.
Introduction
Provide additional background on Eriocheir sinensi, including production, origin, culture status, genes, and so on. And explain why you are studying these two genes on crabs?
Please add some background on PI3K research in aquatic animals to provide context for your paper and capture the reader's attention.
Please describe the significance of this study.
Material and methods
What's the authorization number in line100?
Other genes appearing in Table 1 are also required to provide GenBank numbers.
Table 1 Adding Figure ‘Notes’. Notes: F, forward. R, reverse.
The T7 promoter of the double interference primer used for RNAi should be underlined.
Please add a figure of the experimental design.
I'm confused about the amount of dsRNA you're using. It should be in μg/g, which means that each crab needs to be weighed to determine how much dsRNA to inject, not 100 μL each!
In order to maximize the quality and reliability of this research, I suggest this animal research should comply with all 21 items of the ARRIVE guideline. Please follow the guidelines from the link below: https://arriveguidelines.org/arrive-guidelines
Statistic – please provide information about tests by which Gauss distribution has been evaluated.
Results
Why is Es-KIF2A sometimes lowercase and sometimes uppercase?
Including aquatic animals in line358?
Discussion
Is this the first study of Es-KIF2A and Es-PLK1 in Eriocheir sinensi? If so, then some results are essential to show, including nucleotide and deduced amino acid sequences, alignment of amino acid sequences, phylogenetic tree, etc.
The graphical note to Figure 10, which contains the discussion, needs to be revised.
The first thing to recognize is that your results are very beautiful. However, it is not reasonable that in vitro overexpression experiments were performed using a human cell line, despite the fact that KIF 2A and PLK 1 are very conserved during evolution, and furthermore the results are completely opposite to the in vivo results. You can delete this section or use new cell lines such as shrimp, fish and other aquatic animals. Also I don't see sequence comparison figure for KIF 2A and PLK 1 in the various species.
Comments on the Quality of English LanguageModerate editing of English language required
Author Response
Reviewer 1:
The article “KIF2A upregulates PI3K/AKT signaling through polo-like kinase 1 (PLK1) to affect the proliferation and apoptosis levels of Eriocheir sinensis spermatogenic cells” by Zhao et al. is dealing with a very interesting topic with a vast applied potential. In this study, they knocked down the Es-Kif2a gene by injecting dsRNA into E. sinensis, and overexpressed Es-KIF2A protein in HEK293T cell lines in vitro. Then the cell proliferation level, apoptosis level and PI3K/AKT signaling expression level were detected. Similarly, they inhibited Es-Plk1 gene expression by injecting PLK1 inhibitor BI6727 into E. sinensis, and overexpressed Es-PLK1 protein in HEK293T cell lines in vitro. The cell proliferation level, apoptosis level and PI3K/AKT signaling expression level were also detected. Finally, Western blotting was conducted to explore the upstream and downstream relationship between Es-KIF2A and Es-PLK1. However, some major issues significantly compromised the quality of this MS. I suggest this article be rejected.
Author Response: Thank you so much for these constructive suggestions and questions which largely improve the logical coherence of our review! We have revised the manuscript carefully based on your valuable suggestions.
Abstract:
The first time a species name or other abbreviation must be written in its entirety, and subsequent reoccurrences should be abbreviated. Eriocheir sinensis- (E. sinensis)
Author Response: Thanks for your valuable suggestion. This error has been revised.
Gene names should be italicized.
Author Response: Thanks for your valuable suggestion. This error has been revised.
Introduction:
Provide additional background on Eriocheir sinensis, including production, origin, culture status, genes, and so on. And explain why you are studying these two genes on crabs?
Author Response: Thanks for your valuable suggestion. We have added the background of Eriocheir sinensis to the 1st paragraph of Introduction, and why I choice these two genes to the 2nd~5th paragraph of Introduction.
Please add some background on PI3K research in aquatic animals to provide context for your paper and capture the reader's attention.
Author Response: Thanks for your valuable suggestion. We have added this part to the 3rd paragraph of Introduction.
Please describe the significance of this study.
Author Response: Thanks for your valuable suggestion. We have added this part to the 6th paragraph of Introduction and to the Discussion and Perspectives.
Material and methods:
What's the authorization number in line100?
Author Response: Thanks for your question. We have added the authorization number in line 141.
Other genes appearing in Table 1 are also required to provide GenBank numbers.
Author Response: Thanks for your valuable suggestion. We have added the GenBank numbers of Es-Akt and Es-Caspase-3 in the ‘Note’ (line 165-166), but other genes have not yet been uploaded to NCBI.
Table 1 Adding Figure ‘Notes’. Notes: F, forward. R, reverse.
Author Response: Thanks for your valuable suggestion. We have added the “Notes” in line 165.
The T7 promoter of the double interference primer used for RNAi should be underlined.
Author Response: Thanks for your valuable suggestion. We have underlined the T7 promoter sequence contained in ds-Kif2a-R primer in Table 1, and added its description in the ‘Note’ (line 165).
Please add a figure of the experimental design.
Author Response: Thanks for your valuable suggestion. We have added Figure1 “The overall experimental design of our study.” In the text.
I'm confused about the amount of dsRNA you're using. It should be in μg/g, which means that each crab needs to be weighed to determine how much dsRNA to inject, not 100 μL each!
Author Response: Thanks for your valuable suggestion. These errors have been revised in Materials and methods (2.7 and 2.8). The usage of dsRNA in our study refers to Liu’s study (Liu et al., 2016).
In order to maximize the quality and reliability of this research, I suggest this animal research should comply with all 21 items of the ARRIVE guideline. Please follow the guidelines from the link below: https://arriveguidelines.org/arrive-guidelines.
Author Response: Thanks for your valuable suggestion. We have revised our Material and methods (2.1, 2.7, 2.8) and added limitations of our study in Discussion and Perspectives (4.1, 4.3) according to the ARRIVE guideline.
Statistic-please provide information about tests by which Gauss distribution has been evaluated.
Author Response: Thanks for your valuable suggestion. We have added this information in the figure legends of Figure 3,4,6,7,8.
Results:
Why is Es-KIF2A sometimes lowercase and sometimes uppercase?
Author Response: Thanks for your question. The uppercase form (Es-KIF2A) represents the protein, while the lowercase form (Es-Kif2a) represents the gene.
Including aquatic animals in line358?
Author Response: Thanks for your question. Actually, in addition to trying to culture E. sinensis spermatogonia and hemocytes in vitro, according to previous studies (Liang et al., 2012; Gu et al., 2014), we also tried to culture Drosophila and Danio rerio cell lines in vitro. However, the primary cells we cultured could only survive for a few days, or could not be sub-cultured or used for in vitro transfection experiments. In the future, we will try to cultivate more cell lines from species closely related to E. sinensis, and further optimize cell line culture methods.
Discussion:
Is this the first study of Es-KIF2A and Es-PLK1 in Eriocheir sinensis? If so, then some results are essential to show, including nucleotide and deduced amino acid sequences, alignment of amino acid sequences, phylogenetic tree, etc.
Author Response: Thanks for your valuable suggestion. We have added nucleotide and amino acid sequences, multiple amino acid sequence alignment, phylogenetic tree analysis of Es-Kif2a and Es-Plk1 in the supplementary materials (Fig S1-S4).
The graphical note to Figure 10, which contains the discussion, needs to be revised.
Author Response: Thanks for your valuable suggestion. We have revised the graphical note of Figure 10.
The first thing to recognize is that your results are very beautiful. However, it is not reasonable that in vitro overexpression experiments were performed using a human cell line, despite the fact that KIF2A and PLK1 are very conserved during evolution, and furthermore the results are completely opposite to the in vivo results. You can delete this section or use new cell lines such as shrimp, fish and other aquatic animals. Also, I don't see sequence comparison figure for KIF2A and PLK1 in the various species.
Author Response: Thanks for your valuable suggestion. We have deleted this section. In the future, we will try to generate crab cell lines, or validate all of our data using cell lines from shrimp, fish or other aquatic animals.
Other changes of the paper due to the deletion of in vitro results:
- Simply Summary (line 13-16), Abstract (line 17-35), Discussion and Perspectives (line 556-616) and Conclusion (line 618-621) were all revised.
- Deleted overexpression primers and added Bcl-2 antibody primers in Table 1.
- Deleted the 2nd paragraph of 2.4 (line 164-181), 2.10 (line 247-257), 2.11 (line 259-276), 2.13 (line 285-288) and 2.14 (line 289-292), revised and renumbered 2.12 and 2.15 in Materials and Methods.
- Deleted 3.3 (line 356-377), Figure 3 (line 378-404), 3.7 (line 478-491), Figure 7 (line 492-511), revised the contents and legends of Figure 4, Figure 8 and Figure 10, renumbered all figures in Results.
- Added, deleted or moved some references. All references were renumbered.
References Cited in Author’s Response:
Gu W, Yao W, Zhao Y, Pei S, Jiang C, Meng Q, Wang W, 2014. Establishment of spiroplasma-infected hemocytes as an in vitro laboratory culture model of Chinese mitten crab Eriocheir sinensis. Vet Microbiol. 171(1-2):215-20. DOI: 10.1016/j.vetmic.2014.03.016.
Liang T, Ji H, Du J, Ou J, Li W, Wu T, Meng Q, Gu W, Wang W, 2012. Primary culture of hemocytes from Eriocheir sinensis and their immune effects to the novel crustacean pathogen Spiroplasma eriocheiris. Mol Biol Rep. 39(10):9747-54. DOI: 10.1007/s11033-012-1840-4.
Liu ZQ, Jiang XH, Qi HY, Xiong LW, Qiu GF, 2016. A novel SoxB2 gene is required for maturation of sperm nucleus during spermiogenesis in the Chinese mitten crab, Eriocheir sinensis. Sci Rep. 6:32139. DOI: 10.1038/srep32139

Reviewer 2 Report
Comments and Suggestions for Authors
In the article “KIF2A upregulates PI3K/AKT signaling through polo-like kinase 1 (PLK1) to affect the proliferation and apoptosis levels of 3 Eriocheir sinensis spermatogenic cells” Zhao et al. proposes kinesin-13 family member KIF2A interacts with the serine-threonine kinase Polo-like kinase 1 (PLK1) and synergistically regulate the PI3K/AKT pathway during spermatogenesis. The authors show that overexpression of KIF2A and PLK1 increase cell proliferation, decrease cell apoptosis and knock-down of KIF2A or PLK1 reverse these effects. Furthermore, the authors provide some mechanistic insight and show that PLK1 is a downstream factor or KIF2A. While the findings of the paper provide unique insight of the relationship between KIF2A and PI3K/AKT signaling pathway, it lacks critical data such as subcellular location of KIF2A and PLK1 (spindle/ kinetochore/ spindle pole..) during their interaction, immunoprecipitation showing KIF2A physically interacts with PLK1, interaction sites or domains etc. Additionally, most of the immunofluorescence images reported in the paper are poorly represented and lacks sufficient image resolution and consistent image properties (similar brightness/contrast). Addressing some of the concerns as well as additional comments listed below can strengthen the study provided here.
1. In Fig 1, authors mentioned E. sinensis express KIF2A at different stages of spermatogenesis except mature spermatozoa stage. However, based on the immunofluorescence data, it seems there are pretty significant KIF2A expression (red fluorescence) at mature spermatozoa stage, especially compared to spermatocytes and mid-stage spermatid.
a. A higher magnification immunofluorescence image will be helpful to understand the difference in KIF2A expression at different E. sinensis developmental stages.
b. Also an empty/ negative control image will be helpful to differentiate between KIF2A expression and autofluorescence and background fluorescence
2. In Fig 2C, there is very little difference in CDK2 expression between control and ds-kif2a. Please provide a better western blot data to show reduced CDK2 expression in ds-kif2a. Alternatively, please use a different cell proliferation marker to show reduced cell proliferation after kif2a knock-down.
3. In immunofluorescence images, please choose representative images with similar brightness/contrast. Fig 2B, DAPI intensity in 2nd panel (control, 2nd panel) is stronger than the rest of images. Similarly, in Fig 2D, DAPI intensity is higher in ds-kif2a condition compared to control. Such difference in intensity can make it difficult to understand if reduced Edu expression or increased cy3 expression in ds-kif2a is due to kif2a knock down or due to difference in image intensity.
4. Please provide a rationale for why E. sinensis was chosen as a model animal instead of performing the whole study in cell lines. Is there any specific features/ advantages of using E. sinensis ?
5. Line 374 says “the percentage of apoptotic cells increased significantly (Q3 area) after Es-KIF2A overexpression”. This needs to be changed to “decreased” apoptotic cells after Es-KIF2A overexpression.
6. Please provide a higher magnification/ better quality immunofluorescence images to show co-localization of Es-KIF2A and Es-PLK1. It is not clear from the images provided in Fig 5. If getting good quality images in crabs is difficult, representative images in HEK293T cells will also be fine.
7. In figure 9a1 and 9b1, please indicate conditions for different lanes in western blot. In figure 9b1 and 9b2, add the legends for bar graph.
8. It will be great if author can include some insight about if KIF2A and PLK1 form complex with other proteins to activate PI3K/AKT pathway?
Earlier in the paper authors mentioned KIF2A, PLK1 interaction during mitosis depend on MRN complex. KIF2A and PLK1 also interact with DVL complex to form cilia. Similarly, for PI3K/AKT pathway activation by KIF2A and PLK1, are there other proteins or maybe adapter proteins involved?
Author Response
Reviewer 2:
In the article “KIF2A upregulates PI3K/AKT signaling through polo-like kinase 1 (PLK1) to affect the proliferation and apoptosis levels of Eriocheir sinensis spermatogenic cells” Zhao et al. proposes kinesin-13 family member KIF2A interacts with the serine-threonine kinase Polo-like kinase 1 (PLK1) and synergistically regulate the PI3K/AKT pathway during spermatogenesis. The authors show that overexpression of KIF2A and PLK1 increase cell proliferation, decrease cell apoptosis and knock-down of KIF2A or PLK1 reverse these effects. Furthermore, the authors provide some mechanistic insight and show that PLK1 is a downstream factor or KIF2A. While the findings of the paper provide unique insight of the relationship between KIF2A and PI3K/AKT signaling pathway, it lacks critical data such as subcellular location of KIF2A and PLK1 (spindle/ kinetochore/ spindle pole) during their interaction, immunoprecipitation showing KIF2A physically interacts with PLK1, interaction sites or domains etc. Additionally, most of the immunofluorescence images reported in the paper are poorly represented and lacks sufficient image resolution and consistent image properties (similar brightness/contrast). Addressing some of the concerns as well as additional comments listed below can strengthen the study provided here.
Author Response: Thank you so much for these constructive suggestions and questions which largely improve the logical coherence of our review! We have revised the manuscript carefully based on your valuable suggestions.
- In Fig 1, authors mentioned E. sinensis express KIF2A at different stages of spermatogenesis except mature spermatozoa stage. However, based on the immunofluorescence data, it seems there are pretty significant KIF2A expression (red fluorescence) at mature spermatozoa stage, especially compared to spermatocytes and mid-stage spermatid.
a. A higher magnification immunofluorescence image will be helpful to understand the difference in KIF2A expression at different E. sinensis developmental stages.
b. Also an empty/ negative control image will be helpful to differentiate between KIF2A expression and autofluorescence and background fluorescence.
Author Response: Thanks for your valuable suggestion. We have added a description of the morphology of spermatogenic cells at each stage of E. sinensis spermatogenesis to the 1st paragraph of Introduction. In Fig 1 (now is Fig 2), the red fluorescence signal you see is KIF2A expressed in spermatogonia and spermatocytes. For particularly dim cells, which are cells at different levels of the slice, the distance makes it impossible to tell the cell type apart, although they may look like mature spermatozoa.
- In Fig 2C, there is very little difference in CDK2 expression between control and ds-kif2a. Please provide a better western blot data to show reduced CDK2 expression in ds-kif2a. Alternatively, please use a different cell proliferation marker to show reduced cell proliferation after kif2a knock-down.
Author Response: Thanks for your valuable suggestion. In Fig 2C (now is Fig 3C), it is true that the difference is not obvious from the CDK2 band alone, but the expression level of actin band should also be considered. Moreover, significance analysis showed that the decrease of CDK2 expression was significant. The integrated density ratios (CDK2/actin) in control group are 1.471553398, 1.615396365, 1.115534586, in ds-Kif2a group are 0.69953079, 0.866453127, 1.070249738 (Intensity ratio data for all Western Blot figures are provided in the Supplementary materials).
- In immunofluorescence images, please choose representative images with similar brightness/contrast. Fig 2B, DAPI intensity in 2nd panel (control, 2nd panel) is stronger than the rest of images. Similarly, in Fig 2D, DAPI intensity is higher in ds-kif2a condition compared to control. Such difference in intensity can make it difficult to understand if reduced Edu expression or increased cy3 expression in ds-kif2a is due to kif2a knock down or due to difference in image intensity.
Author Response: Thanks for your valuable suggestion. We have adjusted the brightness /contrast in Fig 2B and Fig 2D (now is Fig 3B and Fig 3D). And we also added the magnification images in Fig 2B (now is Fig 3B).
- Please provide a rationale for why E. sinensis was chosen as a model animal instead of performing the whole study in cell lines. Are there any specific features/ advantages of using E. sinensis?
Author Response: Thanks for your question. E. sinensis is an ideal experimental animal for studying crustacean spermatogenesis (as the description we added in the 1st paragraph of Introduction). We hope to conduct some mechanistic studies in E. sinensis to deepen people's understanding of spermatogenesis of crustacean and species that produce the non-flagellar sperm. In the future, we plan to validate all of our results in cell lines in vitro.
- Line 374 says “the percentage of apoptotic cells increased significantly (Q3 area) after Es-KIF2A overexpression”. This needs to be changed to “decreased” apoptotic cells after Es-KIF2A overexpression.
Author Response: Thanks for your valuable suggestion. At the request of another reviewer, we have deleted this part of the results.
- Please provide a higher magnification/ better quality immunofluorescence image to show co-localization of Es-KIF2A and Es-PLK1. It is not clear from the images provided in Fig 5. If getting good quality images in crabs is difficult, representative images in HEK293T cells will also be fine.
Author Response: Thanks for your valuable suggestion. We have added the higher magnification images in Fig 5.
- In figure 9a1 and 9b1, please indicate conditions for different lanes in western blot. In figure 9b1 and 9b2, add the legends for bar graph.
Author Response: Thanks for your valuable suggestion. We have added the conditions in 9a1 and 9b1 (now is 8a1 and 8b1), and legends for 9a2 and 9b2 (now is 8a2 and 8b2).
- It will be great if author can include some insight about if KIF2A and PLK1 form complex with other proteins to activate PI3K/AKT pathway?
Author Response: Thanks for your valuable suggestion. We added this part in the 3rd paragraph of 4.2 in Discussion and Perspectives and Figure 9.
Earlier in the paper authors mentioned KIF2A, PLK1 interaction during mitosis depend on MRN complex. KIF2A and PLK1 also interact with DVL complex to form cilia. Similarly, for PI3K/AKT pathway activation by KIF2A and PLK1, are there other proteins or maybe adapter proteins involved?
Author Response: Thanks for your question. We added this part in the 3rd paragraph of 4.2 in Discussion and Perspectives and Figure 9.

Round 2
Reviewer 1 Report
Comments and Suggestions for Authors
I have no further comment.
Author Response
Reviewer 1
Comments and Suggestions for Authors
I have no further comment.
Response to Reviewer 1
Thank you for your positive comments
Reviewer 2 Report
Comments and Suggestions for Authors
The authors did a good job of providing rationales for few concerns in the first round, such as, providing a better explanation of why the study is significant, why they chose E. sinensis as model system etc. However, not all the concerns that were mentioned in round 1 were fully addressed. Addressing these couple concerns will be helpful to the final version of the manuscript.
1. Fig 2 (previously fig 1) is still missing negative control. Also authors mentioned in the reply of my comments that “red fluorescence signal you see is KIF2A expressed in spermatogonia and spermatocytes. For particularly dim cells, which are cells at different levels of the slice, the distance makes it impossible to tell the cell type apart, although they may look like mature spermatozoa.” If it is difficult to differentiate the cell types, how the authors differentiated the cell types? Is there any standard biomarkers/ protocols to differentiate cell types at different developmental stages?
2. Authors mentioned low difference in CDK2 expression between control and ds-kif2a is due to difference in actin (loading control) level. If the conclusions from the western blot is not accurate due to difference in loading control, the experiment should be repeated, or a different loading control (tubulin/ GAPDH) should be used.
Author Response
The authors did a good job of providing rationales for few concerns in the first round, such as, providing a better explanation of why the study is significant, why they chose E. sinensis as model system etc. However, not all the concerns that were mentioned in round 1 were fully addressed. Addressing these couple concerns will be helpful to the final version of the manuscript.
Author Response: Thank you so much for these constructive suggestions and questions which largely improve the logical coherence of our review! We have revised the manuscript carefully based on your valuable suggestions.
- Fig 2 (previously fig 1) is still missing negative control. Also, authors mentioned in the reply of my comments that “red fluorescence signal you see is KIF2A expressed in spermatogonia and spermatocytes. For particularly dim cells, which are cells at different levels of the slice, the distance makes it impossible to tell the cell type apart, although they may look like mature spermatozoa.” If it is difficult to differentiate the cell types, how the authors differentiated the cell types? Are there any standard biomarkers/ protocols to differentiate cell types at different developmental stages?
Author Response: Thanks for your question. We have added the negative control (NC) group in Figure 2 and revised the figure legends. Unfortunately, no biomarkers specific to spermatogenic cells have been found in E.sinensis, and that is something we are exploring right now. Actually, we differentiate the cell types at different stages of E.sinensis spermatogenesis mainly by cell morphology, because the morphology of spermatogenic cells varies significantly at different stages (as described in paragraph 1 of Introduction). In mature spermatozoa, most of the cytoplasm is discarded, and the nuclear membrane fits into the plasma membrane, the nucleus becomes cupped (called nuclear cup, NC), enclosing the acrosome structures. In the fluorescence figures, NC can be regarded as the boundary of the mature spermatozoa. Thus, during the stage of mature spermatozoa, if no red fluorescence signal (KIF2A) coincides with NC and its internal structure, it proves that KIF2A is not expressed in this stage.
- Authors mentioned low difference in CDK2 expression between control and ds-kif2a is due to difference in actin (loading control) level. If the conclusions from the western blot is not accurate due to difference in loading control, the experiment should be repeated, or a different loading control (tubulin/ GAPDH) should be used.
Author Response: Thanks for your valuable suggestion. We have replaced the CDK2 data with a better western blot data in Figure 2 (c1 and c3). Reference genes such as GAPDH cannot be found in E.sinensis. Besides, KIF2A is a microtubule depolymerase, so tubulin is not suitable for use as a reference gene in the case of KIF2A knockdown.
